# The Use of Podcasts as Patient Preparation for Hospital Visits—An Interview Study Exploring Patients’ Experiences

**DOI:** 10.3390/ijerph21060746

**Published:** 2024-06-06

**Authors:** Jannie Christina Frølund, Anders Løkke, Hanne Irene Jensen, Ingeborg Farver-Vestergaard

**Affiliations:** 1Department of Medicine, Vejle Hospital, Lillebaelt Hospital, University Hospital of Southern Denmark, Beriderbakken 4, DK 7100 Vejle, Denmark; anders.lokke@rsyd.dk (A.L.); ingeborg.farver-vestergaard@rsyd.dk (I.F.-V.); 2Department of Regional Reseacrh, University of Southern Denmark, J.B. Winsløvsvej 19, DK 5000 Odense, Denmark; hanne.irene.jensen@rsyd.dk; 3Department of Anaesthesiology and Intensive Care, Vejle Hospital, Lillebaelt Hospital, University Hospital of Southern Denmark, Beriderbakken 4, DK 7100 Vejle, Denmark; 4Department of Anaesthesiology and Intensive Care, Kolding Hospital, Lillebaelt Hospital, University Hospital of Southern Denmark, Sygehusvej 24, DK 6000 Kolding, Denmark

**Keywords:** patient involvement, communication, patient information, podcast, qualitative interviews

## Abstract

Introduction: Podcasts have emerged as a promising tool in patient preparation for hospital visits. However, the nuanced experiences of patients who engage with this medium remain underexplored. Objectives: This study explored patients’ experiences of receiving information by way of podcasts prior to their hospital visits. Methods: Semi-structured interviews were conducted with patients with suspected chronic obstructive pulmonary disease (COPD), lung cancer, or sleep apnea. The method of data analysis chosen was thematic analysis. Results: Based on data from 24 interviews, five key themes were identified: technical challenges in utilization of podcasts; individual preferences for information prior to hospital visits; building trust and reducing anxiety through podcasts; the role of podcasts as an accessible and convenient source of information; and enhancement of engagement and empowerment through podcasts. Additionally, the study highlighted the critical importance of tailoring podcasts’ content to individual preferences to optimize the delivery of healthcare information. Conclusions: Podcasts can serve as a meaningful supplement to traditional information sources for patients. However, it is important to recognize that not all patients may be able to engage with this medium effectively due to technical challenges or personal preferences.

## 1. Introduction

Preparation for hospital visits has been found to contribute to reducing the anxiety and fear often associated with medical encounters [1,2], while also assisting healthcare professionals to execute healthcare procedures more easily and promoting patients’ self-management [3]. Patient information materials serve as valuable tools in empowering patients to actively participate in their healthcare decisions and enhance overall health literacy [4,5,6,7]. Patient information is often delivered in written form, e.g., through patient information leaflets or letters [8,9]. However, approximately 16% of the adult Danish population lacks basic reading skills [10]. Individuals with reading and writing difficulties may have less access to relevant health- and treatment-related information, which can contribute to health inequality [10,11]. The European Union (EU) places a significant emphasis on internet accessibility to ensure equal access to websites, applications, and digital services for individuals with disabilities. The EU has issued regulations and directives to promote web accessibility and enhance the online experience for people with disabilities, which aligns with the broader goal of facilitating access to healthcare-related information [12]. While web accessibility typically focuses on text-based content, alternative formats, such as podcasts, can also make information more accessible to patients.

Digital media, particularly podcasts, digital audio files available online [13], have attracted interest among medical education providers for use in educational programs [14,15,16] and among healthcare professionals who prepare patients for hospital visits [17]. Patients with disabilities or impairments often face challenges in accessing traditional patient information materials, such as brochures and pamphlets that are related to mobility limitations, visual or hearing impairments, or challenges in reading comprehension [18,19,20]. Podcasts present a valuable alternative format for some of these individuals. Podcasts can easily be accessed through the internet and are compatible with a wide range of devices, making them convenient and flexible for users. Furthermore, podcasts have the potential to accommodate different learning styles [16] and enhance health literacy and knowledge dissemination in educational settings [21].

The utilization of podcasts in health communication is relatively new [16,22]. Prior to conducting the present study, we developed a user-centered podcast series [17,23,24,25], aimed at assisting patients in preparing for hospital visits. Using the ‘empathy map’ method, we envisioned patients as the target audience of the podcasts, considering their potential information needs prior to hospital visits [26,27,28]. However, formalized insight into patients’ experiences of podcasts is currently lacking [17]. Hence, there is a need for a deeper understanding of patients’ experiences with the use of podcasts for patient information, in terms of impact, accessibility, and perceptions. It is essential to gain insights into patients’ thoughts and experiences in order to tailor and enhance the quality of these resources and to better align them with the diverse needs of patients.

Based on the above, the objectives of this interview study were to determine: (1) patients’ experiences of receiving information through podcasts, and (2) the patients’ views on podcasts, including their usefulness, their format, and comprehensibility.

## 2. Materials and Methods

### 2.1. Study Design

We conducted a qualitative study based on semi-structured interviews with patients suspected of having lung cancer, chronic obstructive pulmonary disease (COPD), or sleep apnea.

### 2.2. Participants

Twenty-four participants were recruited from three separate outpatient clinics at a Danish hospital: a lung cancer diagnostic workup, a COPD clinic, and a sleep apnea clinic. To capture a broad range of patients’ experiences from both chronic patients and patients with cancer, we recruited eligible participants who met the following criteria:Referral to the hospital because of suspected lung cancer, COPD, or sleep apnea;≥18 years old;Able to understand and speak Danish.

Participants with severe mental illness, severe cognitive disability, or terminal illness were excluded.

Participants were sampled across genders, ages, and suspected diagnoses. An equal distribution was sought between patients who had heard the podcast and those who had not, to explore both perspectives. Patients could choose their preferred interview setting at either the hospital, via telephone, or at their home. This flexible approach aimed to accommodate diverse needs and preferences and to take the patients’ comfort, convenience, and accessibility into consideration.

### 2.3. Podcasts

In April 2021, three different user-centered podcasts [23,24,25], designed to prepare patients with suspected lung cancer, COPD, and sleep apnea for hospital visits, were introduced and made available for patients’ access. These podcasts were created through collaboration with patients undergoing diagnosis, ensuring their practical relevance. The podcasts were structured in a dialogue-based format, presenting conversations between the patients and clinicians. The podcasts are divided into various sections, such as the podcast’s introduction and sections that address questions such as “Why have I been referred to the hospital?” and “What will happen during my first visit?” They also cover the progress of examinations for the three different diseases and what patients can do while awaiting the results. Moreover, there is a summary of the podcast’s content, outlining key information, such as the procedures, expectations, and resources available. A professional podcast agency handled the recording and editing processes A detailed description of the design and development of the podcasts can be found in another article [17].

All patients referred to the clinic received a URL linking to the podcast as part of their welcome package before their scheduled hospital visit. The podcast for each patient was chosen based on their referral from the general practitioner, ensuring that it was tailored to their specific diagnosis. While it was strongly recommended to listen to the podcast, doing so was not mandatory.

### 2.4. Data Collection

A semi-structured interview guide (outlined in Table 1) was developed by the research team, based on the project’s objectives and the existing literature [29,30,31]. The interview guide consisted of open-ended and follow-up questions [32], addressing various topics such as the participants’ backgrounds; reasons for referral to the hospital; their general experiences with patient information, specifically regarding the podcasts for patient information; and whether they had listened to the podcast. The interview guide intentionally combined disease literacy and podcast-related questions. Questions at the beginning and at the end eased participants into the conversation for a more comfortable interview experience.

The 24 interviews were conducted by a psychologist with a PhD and a nurse with an MA, both with prior qualitative research training and expertise. They did not know the participants and were not involved in their care. The first interview was a pilot interview to evaluate the utility of the interview guide. An analysis of the results of the pilot indicated that no revisions to the interview guide were required, which suggested that the pilot interview was executed effectively. As a result, data obtained from the pilot interview were incorporated into the comprehensive analysis. All interviews were audio-recorded and transcribed verbatim by the first author.

### 2.5. Data Analysis

Thematic analysis [33] was used in this study to gain a nuanced understanding of patients’ experiences and perspectives of engaging with podcasts to receive patient information. The thematic analysis followed six phases, as outlined in Table 2 [33]. To enhance the reliability, the identified themes were repeatedly discussed by two authors, with a view to improving their validity and providing a more nuanced analysis.

### 2.6. Ethics

The Regional Committee on Health Research Ethics for Southern Denmark was informed about the study and determined that the study required no formal ethical approval (20202000-222). The study was registered at the Regional Danish Data Protection Agency (20/55212).

Informed consent was obtained from all participants before the interviews, with meticulous attention given to guaranteeing their voluntary engagement, confidentiality, and anonymity. The departmental staff, led by the nurses, informed the patients about the interview study during an outpatient visit and distributed written information and a consent letter for the participants to sign. Participants were given 24 h of time for consideration. Both oral and written information about the study were provided. Confidentiality and the freedom to withdraw from the study at any time without any impact on treatment or care were emphasized. During the interviews, the participants were additionally informed that the interviewer had no involvement in their treatment and therefore was not in a position to answer questions related to it. Pseudonyms or participant IDs were used to protect their identities in any reported findings or publications.

## 3. Results

Twenty-four patients with suspected lung cancer, COPD, or sleep apnea were interviewed. Details of the participants’ characteristics are presented in Table 3. Additionally, five patients declined to participate because of illness or low energy levels. The patient cohort was evenly divided, with half having listened to a designated podcast before their hospital appointment, while the other half had not. Telephone interviews served as the primary mode of communication for all patients, except for one, who opted for an in-person interview during their hospital visit. The interviews occurred between May 2021 and November 2023, typically within a week of the hospital appointment. The interviews’ durations ranged from 11 to 37 min.

The following themes were identified:Objective 1: Patients’ experiences of receiving information through podcasts.○Theme 1: Technical challenges in utilization of the podcasts for patient preparation. ○Theme 2: Individual preferences for information prior to hospital visits.○Theme 3: Building trust and reducing anxiety through podcasts.
Objective 2: Patients’ views on podcasts, including their usefulness, format, and comprehensibility.○Theme 4: Podcasts as an accessible and convenient source of information.○Theme 5: Enhanced engagement and empowerment through podcasts.

### 3.1. Theme 1: Technical Challenges in Utilization of the Podcasts for Patient Preparation

Some participants expressed that they felt too old to engage with modern digital communication media, such as podcasts, stating that podcasts are tailored exclusively to the younger generation.


*Podcasts are probably more suited to the younger generation. Personally, I feel I’m way too old to engage with such modern mediums*
(P9).

However, this view was different from other older patients, who argued that the decision to engage with podcasts transcends age and hinges more on individual preferences. They emphasized the value of podcasts as versatile resources for both information and entertainment, asserting that their utility is contingent upon personal choice rather than age-related stereotypes. One participant exemplified this viewpoint, stating,


*I find podcasts to be a valuable resource for information and entertainment. It’s not about age; it’s about personal choice*
(P21).

In addition to attitudinal barriers, technical obstacles emerged as an impediment to the effective utilization of podcasts for patient preparation. Notably, issues related to internet connectivity were prevalent among participants, with some encountering difficulties in accessing or streaming podcast episodes. These technical challenges frustrated participants, and they highlighted the potential limitations of digital resources, such as podcasts, when reliable internet access was not assured.


*I faced difficulties accessing the podcast episodes, which was frustrating. Technical issues like these can significantly reduce the usefulness of such resources. Nevertheless, I would have listened if I had been able to access it*
(P6).

Moreover, gaps in communication and awareness were apparent; some participants were oblivious to the existence of the podcast altogether. Participants mentioned that this lack of awareness points to potential shortcomings in disseminating information about the available resources, indicating a need for more proactive communication strategies to ensure patients’ engagement. For instance, one participant mentioned, 


*I simply received a hospital appointment, and everything went from there. I wasn’t aware there was a podcast*
(P14).

Furthermore, the interview findings brought to light disparities in digital access and proficiency among participants, particularly those with limited technological skills. For these individuals, navigating podcast platforms posed significant challenges, leading to feelings of exclusion and frustration. Despite these hurdles, some patients demonstrated a strong commitment to overcome technical obstacles, exemplifying their determination to leverage the available resources for better preparedness.


*I tried accessing the podcasts on my iPad, computer, and phone. I even sought assistance from the hospital staff, because I really wanted to understand and be well-prepared*
(P1).

This determination underscored their willingness to make an effort to engage with all the informational resources available.

### 3.2. Theme 2: Individual Preferences for Information Prior to Hospital Visits

The interviews revealed variations in how patients prepare for hospital visits, highlighting individual preferences and reactions. Some patients actively seek information and prepare for the visit, while others adopt a more relaxed approach without feeling the need for prior preparation. One patient expressed a lack of interest in the podcast material, preferring to approach the visit without extensive research. 


*I don’t think a podcast like that would interest me at all. I haven’t really listened to it much or read about it extensively. I simply take it as it comes without expecting anything serious from it*
(P5).

Conversely, another patient found the podcast instrumental in their preparation, noting heightened confidence about the visit.


*I felt fully prepared for the hospital visit, knowing exactly what to expect. I must say, the visit unfolded exactly as described in the podcast*
(P7).

Patients exhibited diverse experiences in their approach to preparing for their hospital appointments. Some individuals felt adequately informed with basic details but lacked insight into the visit’s specifics. Conversely, others, who had read written patient information, felt fully equipped for the appointment, a sentiment particularly influenced by limited energy levels. For instance, one patient expressed feeling adequately prepared solely with the knowledge of the appointment’s date and time, while another patient felt sufficiently informed without a wish for further information, especially because their energy level was low. 


*I felt completely informed, leaving me with no desire for further information, especially considering my depleted energy reserves*
(P18).

However, among those who had not accessed the podcast material, there was a notable expression of uncertainty and, in some instances, anxiety regarding their impending hospital visit. This lack of clarity and apprehension might not stem solely from not having engaged with the material but could also be attributed to the inherently dynamic nature of hospital visits. With rapid changes and a multitude of considerations, accessing and comprehending relevant resources may become challenging for some individuals. 


*I was really confused and concerned because of the sudden rush in which I was referred for the examinations. The lack of information made it even more challenging. I felt a bit lost and overwhelmed by the situation*
(P13).

The patients’ responses exhibited notable diversity based on their individual conditions and the information presented through the podcast. They reflected on how this awareness transformed their initially nonchalant attitude, leading to a sense of informed calmness rather than uncertainty. Among them, several patients experienced a notable shift in attitude towards their symptoms and disease. For instance, one patient expressed a newfound understanding of the importance of medical assessments after engaging with the podcast material. 


*I became aware of why the assessment is so important. It was surprising that I still maintained my nonchalant attitude. It felt more like an informed calmness than uncertainty. This has truly taught me that even if you’re not worried, it doesn’t mean you shouldn’t be informed*
(P20).

### 3.3. Theme 3: Building Trust and Reducing Anxiety through Podcasts

The majority of participants who had used the podcasts emphasized the important role of podcasts in building trust and reducing anxiety related to their healthcare experiences. Their narratives shed light on how podcasts served as a valuable tool in the enhancement of their confidence and in easing apprehensions. For many, listening to the podcasts provided a sense of reassurance. Hearing insights from healthcare professionals and fellow patients about the hospital visit and what to expect played a crucial role in building trust in the information they received. Participants found that the podcasts were personalized and targeted their specific needs:


*I felt reassured listening to the podcasts. Hearing from healthcare professionals and other patients helped build trust in the information provided. It felt like a personalized conversation tailored to my needs*
(P2).

In terms of reducing anxiety, several participants found that tuning into the podcasts helped alleviate some of the stress associated with the upcoming hospital visits. They felt better prepared and equipped to navigate their healthcare experiences, leading to a significant reduction in their stress levels.


*Listening to the podcasts helped alleviate some of my anxiety about the upcoming hospital visit. I felt more prepared and knew what to expect, which significantly reduced my stress levels*
(P3).

However, even participants who had not listened to the podcasts echoed similar sentiments of feeling reassured and confident in their healthcare encounters, indicating a general reduction in anxiety levels when adequately prepared for hospital appointments. They highlighted aspects such as the competence of the healthcare team and the information provided before their hospital visits, all of which contributed to their sense of ease.


*Feeling that one is in good hands not only alleviates anxiety but also fosters a sense of confidence and security*
(P15).

### 3.4. Theme 4: Podcasts as an Accessible and Convenient Source of Information

Several participants consistently highlighted their appreciation for the accessibility and convenience provided by podcasts. They expressed delight in the flexibility afforded by podcasts, allowing them to tune in at their convenience, be it during their daily commute, while attending to household chores, or even during leisure time spent with family members. The ability to seamlessly integrate consumption of the podcasts into various facets of their lives was noted as an advantage over other formats of information delivery.


*I like that I can listen to podcasts whenever I want, whether it’s during my commute, while I’m doing chores at home, or together with my wife*
(P8).

Several participants emphasized the convenience of podcasts as an alternative source of information. They mentioned that being able to access podcasts across different platforms, such as phones, tablets, or computers, was highly convenient. This accessibility was considered essential in enhancing their engagement with healthcare-related content. Furthermore, participants valued the interactive nature of podcasts, particularly the ability to pause, rewind, and revisit episodes at their own pace. This feature was seen as invaluable in aiding comprehension and retention, allowing listeners to thoroughly digest the information presented. Moreover, several participants noted that podcasts could serve as an effective tool to disseminate healthcare information to patients’ family members or caregivers, thereby fostering better-informed decision making and support networks. Another aspect appreciated by the participants was the simplification of complex information facilitated by podcasts. One participant remarked, 


*Podcasts present information in a way that’s easy to understand. It’s like having a conversation with someone who knows what they’re talking about*
(P7).

The podcasts’ tailored approach in distilling intricate concepts into easily understandable information was perceived as highly advantageous, especially for individuals navigating through medical terminology or managing complex health conditions. Additionally, a majority of participants emphasized the importance of the podcast host’s voice in enhancing their listening experience. A clear and pleasant delivery was deemed important, with one participant stating, 


*The voice of the podcast host is crucial; a pleasant and clear delivery enhances the overall experience, making it more engaging and enjoyable to listen to*
(P12).

This aspect not only contributed to the accessibility of the content but also fostered a sense of connection and engagement with the material presented.

### 3.5. Theme 5: Enhanced Engagement and Empowerment through Podcasts

Several participants emphasized how podcasts enhanced their engagement and empowerment by providing comprehensive information, fostering realistic expectations, and promoting a sense of empowerment. 


*The podcasts provided me with valuable insights into my condition and the upcoming procedure*
(P10).

This sentiment was echoed by others, who highlighted that through engaging with the podcasts and listening to the experiences shared by both fellow patients and healthcare professionals served as a valuable resource to obtain in-depth knowledge about their medical condition and forthcoming treatments. Moreover, several participants described feeling empowered after engaging with podcasts, stating that it made them more proactive in their healthcare decisions. Additionally, they felt more informed about and involved in their healthcare options, attributing this to the insights gained from the podcasts.


*The podcasts empowered me to be more proactive in my treatment decisions and to ask questions, as recommended*
(P22).

Furthermore, several participants appreciated the sense of control and confidence instilled by the podcasts, which encouraged them to explore various treatment options and engage more actively in discussions with their healthcare providers. As one participant articulated,


*Listening to the podcasts made me feel more informed and involved. I felt better equipped to understand my condition and explore different treatment options*
(P2).

## 4. Discussion

Although podcasts have gained increasing interest in the context of utilization of healthcare over the last years, this is the first study, to our knowledge, to uncover patients’ experiences of receiving information through podcasts prior to hospital visits. We found that both technical aspects and the individuals’ preferences and need for information influenced utilization of the podcast. Those who listened to the podcasts generally experienced them as a means of building trust and reducing anxiety before hospital visits. Moreover, we found that podcasts were perceived as an accessible and convenient source of information that enhanced feelings of engagement and empowerment among the users. 

### 4.1. Technical Challenges and Individuals’ Information Preferences

Some participants exhibited diverse attitudes toward podcasts, with older individuals often feeling marginalized by modern digital platforms, perceiving them as being tailored exclusively to a younger audience. However, an alternative perspective was raised by other participants, who highlighted that personal choice was the driving force behind engagement with the podcast, irrespective of age-related appropriateness. These observations resonate with the extant literature, which elucidated patients’ varying understanding of age-related perceptions and technical challenges in healthcare communication [13,34]. Based on these findings, rebranding “podcast” to “audio” might influence older patients’ perceptions, making it more accessible and less associated with modern digital culture. This change could potentially encourage broader engagement and address the digital divide in patient education.

In our study, a paradox emerged concerning the integration of podcast links within written materials. On the one hand, we found that podcasts could serve as a more understandable alternative for individuals who struggle to read written material. On the other hand, patients could only access the online podcast via a URL link or QR code in the written information material. Hence, patients who needed information in an audio format still needed to read written information to access the audio information. This paradox underscores the challenge of low literacy and health literacy, exacerbating communication barriers, particularly among cohorts with a low socioeconomic status [35,36,37]. These challenges can be mitigated by implementing strategies to help patients download the podcast automatically, thereby enhancing accessibility and ensuring that all patients can benefit from the information provided. Despite the potential of podcasts to bridge informational gaps by delivering content in an accessible spoken format [38,39], interviews with participants revealed accessibility hurdles. This emphasizes the general imperative of ensuring universal access and providing the necessary assistance, as highlighted by recent literature [40,41]. Research has suggested that older adults’ positive perceptions of digitalization correlate with their digital literacy skills [42], highlighting the importance of enhancing such skills to foster a more favorable view of the adoption of technology. Hence, our study’s findings closely align with the existing literature, underscoring the need to tailor communication strategies to meet diverse patients’ needs [43,44]. While some patients proactively sought information, others relied on healthcare providers for guidance, and a third group may not seek information at all because of a fear of being confronted with disturbing facts about their health. These findings resonate with the literature [43], and this highlights the need for improved communication between healthcare providers and patients, as well as acknowledging patients’ preferences regarding the level of information. Engaging with informational resources such as podcasts can provide reassurance and equip individuals with knowledge about what to expect during healthcare encounters. This engagement can catalyze a shift in patients’ perspectives, empowering them to make informed decisions about their healthcare journey [45].

### 4.2. The Podcasts’ Format and Accessibility

Our findings further support the notion that—for those who know they exist and can use them—podcasts can serve as a convenient and easily accessible source of healthcare information, in line with previous research [38,39]. The participants in our study emphasized the flexibility of podcast consumption, noting its convenience across different platforms and during daily activities.

Our study contributes to the understanding of the role of podcasts in healthcare communication, revealing their impact on building trust and reducing anxiety among healthcare consumers. This finding echoes previous research that has showcased the efficacy of informational interventions in healthcare settings, particularly in alleviating anxiety related to medical procedures [40]. Such effectiveness could be attributed to various factors inherent in podcasts, including the comforting presence of a human voice, the use of everyday language rather than complex written terms, and the accessibility of information. These factors collectively contribute to a sense of reassurance and comprehension among listeners [15,46].

Moreover, our study shed light on the broader implications of healthcare communication initiatives. Participants who had not actively engaged with the podcasts still expressed similar sentiments of reassurance and confidence. This underscores the benefit of accessible and relatable healthcare information and its role in fostering patients’ trust and in reducing anxiety across diverse populations. It also demonstrates that podcasts are one among many other means of communicating with patients and could be regarded as a supplement, rather than an alternative, to other forms of communication materials.

In the present study, the participants highlighted that engagement with the podcasts led to increased proactivity in healthcare decision-making and a heightened sense of confidence and control. Participants reported feeling more informed and involved in their healthcare options, which empowered them to advocate for themselves and actively participate in their treatment plans. Studies have shown that patient empowerment starts with information and education, enabling patients to actively participate in treatment decisions [41]. Additionally, podcasts hold promise as a tool to support shared decision-making [47] by providing easily accessible information and bolstering patients’ engagement in healthcare decisions. This integrated approach could prove beneficial in improving patient-perceived quality and treatment outcomes across diverse medical conditions. This is consistent with previous research [48] on effective healthcare communication, which recommends utilizing diverse formats, such as audiovisual, audio, and written healthcare information. To reduce cognitive demand, it is recommended that certain strategies should be used, such as basic design, lay language, glossaries, simple narration, visual reinforcement, summaries, minimal on-screen text, simplified content, contextual information before factual details, and usage of the active voice.

### 4.3. Strengths and Limitations

One notable strength of the current study lies in the utilization of semi-structured interviews in a group of patients from three different respiratory clinics, which enhanced the breadth of experiences of podcasts in the hospital setting. The patients were interviewed soon after their decision to participate, which decreased the risk of memory errors. However, it is important to acknowledge the limitations inherent in semi-structured interviews, including the fact that participants are those who voluntarily agree to participate. Notably, five patients declined to participate due to illness or a lack of energy, potentially impacting the richness and diversity of the collected data. The group of participants in the present study included patients with suspected lung cancer, COPD, and sleep apnea, which are predominantly conditions seen in older individuals. Younger participants may have other experiences of podcasts in this context, but this was outside the scope of this particular study to address that research question. While most participants favored telephone interviews, one participant opted for an in-person interview at the hospital. Although non-verbal cues cannot be captured in remote interviews, telephone interviews were deemed to be appropriate because they offered patient convenience.

The findings from this study, as in other qualitative studies, do not claim to be generalizable. However, the results of this study should be transferable to other healthcare settings and patient populations, particularly those considering the integration of podcasts as informational resources.

### 4.4. Implications for Practice

Our study highlights the significance of podcasts in augmenting patients’ knowledge and readiness ahead of hospital visits. Podcasts not only serve as informational resources but they also provide emotional support and foster connections between patients and healthcare professionals. However, the effectiveness of podcasts hinges on their specific content and the relevance of the contents to the patients’ information needs. The content of the podcasts in the present study was developed on the basis of a comprehensive, user-based process [17]. Hence, it is important to recognize that the participants’ experiences in the present study cannot be directly extrapolated to other podcasts or other settings.

Patients’ information needs vary depending on their medical conditions, underscoring the need for customized content. In light of our findings, healthcare providers should recognize the diverse needs and preferences of patient populations, especially regarding age-related perceptions and technological barriers. By aligning communication strategies with patients’ preferences and capabilities, healthcare providers can optimize the effectiveness of healthcare communication and improve patients’ engagement and outcomes. 

As podcasting continues to evolve, it presents healthcare professionals and organizations with an alternative avenue toward the improvement of disseminating patient information. Embracing this medium can result in more personalized and accessible patient information, ultimately enhancing the delivery of healthcare. However, larger trials are needed in order to test the effectiveness of podcasts as preparatory tools for hospital visits, focusing on their potential to reduce anxiety and empower individuals in making informed healthcare decisions.

## 5. Conclusions

Our interview study contributes insights into the diverse experiences of patients who engage with podcast-based patient information. Understanding patients’ perspectives can inform the development and optimization of implementing podcasts, ultimately improving patients’ hospital visit experiences. We found that podcasts, influenced by technical factors and individual preferences, were valued by some patients as a trusted resource. Those who engaged with podcasts reported benefits such as reduced anxiety, increased trust, and enhanced empowerment. Overall, podcasts are a valuable addition to the existing information strategies and offer a convenient platform for patients to access relevant information, potentially transforming healthcare-related preparation and improving overall experiences.

## Figures and Tables

**Table 1 ijerph-21-00746-t001:** Interview guide.

Theme	General Questions	Clarifying Questions
Introduction	1. Please provide a brief summary of your background and qualifications.	1a. Age, education, job, family/children
Life pre-hospital admission	2. What symptoms have you experienced?	2a. For how long have you had the symptoms? 2b. How were you referred to the hospital?
	3. How do your symptoms generally impact your everyday life?	
The conversation	4. When you were informed about commencing the process at the hospital, what were your initial thoughts?	4a. What were your emotions regarding this?
	5. What went through your mind when you discovered there was a podcast available to listen to before your first visit?	5a. The letter * containing the link to the podcast?
	6. Did you contemplate whether or not to listen to it?	6a. At what point did you listen to the podcast? 6b. Where were you when you listened to the podcast? 6c. Have your spouse or other relatives also listened to the podcast?
	Note: If the person had not listened to the podcast:	Note: Considering the reasons for not listening to the podcast, proceed to Question 10.
	7. What thoughts or emotions did the idea of the podcast evoke?	7a. How did you perceive the information presented in the podcast? 7b. Did anything catch you by surprise? 7c. How did you find the duration of the podcast?
	8. How do you feel the content of the podcast aligned with your initial experiences upon entering the hospital for the first time?	8a Did you feel adequately prepared for your initial visit?
	9. Have you taken any actions or followed up on anything after listening to the podcast?	9a. For instance, prior to your first visit? 9b Is there anything specific you wished to know more about?
Future	10. Was there anything you missed or needed to feel well-prepared for in the process?	10a. Is there anything the hospital could have done differently? 10b. If so, how?
Rounding off the interview	11. Is there anything we have not covered that you feel is important to mention? 12. How has the experience of participating in this interview been for you?	

* The letter was sent via e-boks, a Danish digital mailbox service.

**Table 2 ijerph-21-00746-t002:** Thematic analysis in six phases.

Phase	Description
Familiarization	Transcribing data and thoroughly reading the qualitative material to gain an overall understanding, identifying the initial impressions, patterns, or ideas.
2. Generating initial codes	Systematically coding the data to label and categorize meaningful units of information, using an inductive approach to allow the codes to emerge directly from the data.
3. Searching for themes	Organizing the codes into potential themes based on their conceptual relevance and their relationship to the research questions, representing recurring patterns, ideas, or concepts within the data.
4. Reviewing themes	Interpreting and analyzing the identified themes to extract meaningful insights, examining the relationships between themes, exploring deviant cases, and considering broader contexts for a comprehensive understanding.
5. Defining and naming themes	Ongoing analysis to refine the specifics of each theme and overall story, generating clear definitions and names for each theme.
6. Producing the report	Selecting representative quotes or excerpts from the data to illustrate and support each theme, enhancing the transparency and credibility of the process of analysis.

**Table 3 ijerph-21-00746-t003:** Characteristics of the interviewed participants (N = 24).

Participant	Age	Sex Female (F) Male (M)	Social Status	Work Status	Had the Participant Heard the Podcast?
Suspected lung cancer					
P1	66	F	Married	Retired	No
P2	72	M	Married	Part time employed	Yes
P3	65	F	Widowed	Disibility pension	Yes
P4	63	M	Married	Retired	No
P5	51	M	Single	Disibility pension	No
P6	83	M	Married	Retired	No
P7	65	M	Single	Retired	Yes
P8	65	M	Married	Retired	Yes
Suspected COPD or COPD					
P9	83	M	Married	Retired	No
P10	76	M	Married	Retired	Yes
P11	76	F	Married	Retired	Yes
P12	64	F	Single	Disibility pension	Yes
P13	62	M	Married	Retired	No
P14	75	M	Married	Retired	No
P15	79	F	Single	Retired	No
P16	73	F	Married	Retired	Yes
Suspected sleep apnea					
P17	44	F	Married	Employed	No
P18	57	F	Married	Unemployed	No
P19	58	F	Married	Employed	No
P20	52	F	Married	Employed	Yes
P21	85	F	Married	Retired	Yes
P22	57	F	Divorced	Employed	Yes
P23	51	F	Divorced	Employed	Yes
P24	62	M	Married	Employed	No

## Data Availability

The dataset used in the current study is available from the corresponding author on reasonable request.

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
