# Peer review of "The Use of Podcasts as Patient Preparation for Hospital Visits—An Interview Study Exploring Patients’ Experiences"

_ijerph, 2024, doi:10.3390/ijerph21060746_

Round 1

Reviewer 1 Report

Comments and Suggestions for Authors

Dear authors, I have read and evaluated the paper entitled “The Use of Podcasts as Patient Preparation for Hospital visits – 2 An Interview study Exploring Patients’ experiences”. The topic you have chosen, exploring the use of podcasts in patient preparation, is not only intriguing but also holds significant potential in improving healthcare experiences.

While reading your paper, I found myself pondering over a few points that I believe could enhance the depth and clarity of your research.

1.      Maybe it could help to explain the podcast in the introduction as a form of communication and its difference from other types. The advantage of a podcast is not only available through the Internet but also on devices, so the users can download and listen to them at any time. The problem that you mention in the study (that people should still read if it is a link in the letter from the hospital) can be solved by helping patients download the podcast automatically.

2.      In terms of the research design, could you provide more clarity on the strategy of choosing patients? For instance, did you aim to compare their perception of a podcast depending on their diagnosis? If not, why were these three categories chosen? It would be interesting to know if the podcast had an impact. Additionally, did you ask patients (not mentioned in the interview protocol) if they had prepared themselves in other ways for their hospital visits? Was listening to the podcast mandatory, considering that some written information is mandatory for patients to read before their hospital visits? Was the podcast specific to the diagnosis? Who were the animators of the podcast? You mentioned that more details can be found in another article, but could you also provide them here for a more comprehensive understanding of your research?

3.      There was some age bias as the majority were retired and the youngest participant was 44, so patients perceived that podcasts were suitable for younger people. Do you think if you change the word “podcast” to “audio” aged people could perceive it differently?  I think you can discuss it in the practical propositions.

Author Response

Reviewer: 1

  • Comment: Dear authors, I have read and evaluated the paper entitled “The Use of Podcasts as Patient Preparation for Hospital visits – An Interview study Exploring Patients’ experiences”. The topic you have chosen, exploring the use of podcasts in patient preparation, is not only intriguing but also holds significant potential in improving healthcare experiences.
    • We appreciate the positive comments of the reviewer.
  • Comment: Maybe it could help to explain the podcast in the introduction as a form of communication and its difference from other types. The advantage of a podcast is not only available through the Internet but also on devices, so the users can download and listen to them at any time. The problem that you mention in the study (that people should still read if it is a link in the letter from the hospital) can be solved by helping patients download the podcast automatically.
    • We have revised the introduction to include a detailed explanation of what a podcast is and how it functions as a unique form of communication. Specifically, we have highlighted the advantages of podcasts, such as their accessibility via the Internet and their availability on various devices, allowing users to download and listen to them at their convenience. Additionally, we have clarified the distinctions between podcasts and traditional forms of information dissemination. This expanded explanation aims to provide a clearer understanding of the medium and its benefits to the audience. Line 60-73.
    • We would like to thank the reviewer for their insightful suggestion regarding the issue in our study, where patients still need to read written material to access podcast links provided in hospital letters. In response, we have added the following: "These challenges can be mitigated by implementing strategies to help patients download the podcast automatically, thereby enhancing accessibility and ensuring that all patients can benefit from the provided information." This addition addresses the problem and highlights a practical solution to enhance accessibility for all patients. Line 408-410.
  • Comment: In terms of the research design, could you provide more clarity on the strategy of choosing patients? For instance, did you aim to compare their perception of a podcast depending on their diagnosis? If not, why were these three categories chosen?
    • In response to your comments, we have revised the Method section to emphasize our aim of capturing a broad range of patient experiences regarding the media podcast as a source of patient information. Therefore, we added the following sentence for clarity: "To capture a broad range of patient experiences from both chronic patients and patients with cancer, we recruited eligible participants who met the following criteria:" (Line 98-100).

This adjustment underscores our intent to include a diverse sample of patients to provide comprehensive insights into the use and impact of the podcast as a source of patient information. We believe this revision better communicates the scope and inclusivity of our study.

  • Comment: It would be interesting to know if the podcast had an impact. Additionally, did you ask patients (not mentioned in the interview protocol) if they had prepared themselves in other ways for their hospital visits?
    • Thank you for pointing that out. Indeed, we did include questions related to patients' preparation for hospital visits in our interview protocol. Specifically, in questions 10 and 10a, we inquired about patients' preparations for their appointments in ways beyond listening to the podcast. This allowed us to gain insights into the various methods patients employed to ready themselves for their hospital visits, thereby providing a more comprehensive understanding of their preparation strategies. We have addressed this aspect under Theme 2 (line 247-290) of our study. During the interviews, patients naturally discussed various ways they prepared themselves for their hospital appointments, including accessing written patient information and relying on their own prior knowledge and experiences. These insights provided valuable context to understand the broader spectrum of patient preparation beyond the podcast alone.
  • Comment: Was listening to the podcast mandatory, considering that some written information is mandatory for patients to read before their hospital visits?
    • While we highly recommend patients to engage with the material provided, including the podcast, it is not feasible to enforce mandatory participation. Therefore we have added: “All patients referred to the clinic received a URL linking to the podcast as part of their welcome package before their scheduled hospital visit. While it was strongly recommended to listen to the podcast, doing so was not mandatory.” (line 132-133).
  • Comment: Was the podcast specific to the diagnosis? You mentioned that more details can be found in another article, but could you also provide them here for a more comprehensive understanding of your research?
    • We have revised the content to clarify that three distinct podcasts were developed. “In April 2021, three different user-centered podcasts [23–25], designed to prepare patients with suspected lung cancer, COPD, and sleep apnea for hospital visits, were introduced and made available for patient access. These podcasts were created through collaboration with patients undergoing diagnosis, ensuring their practical relevance. The podcasts were structured in a dialogue-based format, presenting conversations between patients and clinicians” (line 114-120). (line 125-126).
  • Comment: Who were animators of the podcast?
    • We have added: “A professional podcast agency handled the recording and editing processes.”
  • Comment: There was some age bias as the majority were retired and the youngest participant was 44, so patients perceived that podcasts were suitable for younger people. Do you think if you change the word “podcast” to “audio” aged people could perceive it differently?  I think you can discuss it in the practical propositions.
    • In response to your comment, we have added a line in the discussion: “Based on these findings, rebranding "podcast" to "audio" might influence older patients' perceptions, making it more accessible and less associated with modern digital culture. This change could potentially encourage broader engagement and address the digital divide in patient education.” (line 396-399).

Reviewer 2 Report

Comments and Suggestions for Authors

This paper tends to get insight into the impacts of podcast on the level of patient preparation for a doctor visit.

Despite the smal set of participant, the study assessed all relevant aspects of the topic.

Though, the paper contains some weaknesses which need to be fixed:

1. the abstact lack the study results and conclusion.

2. It is confusing if the podcast is based on a previous diagnosis or only on the symptoms reported by the paptient self? The authors are advised to make this clear.

3. remaing comments: see attached file. You need to put in brackets at some parts of papers  link to the results section. Many of my questions get answered in the discussion and results section. To make the paper easily readable, please the link pointed parts to the results and discussions.

Author Response

Reviewer: 2

  • Comment: This paper tends to get insight into the impacts of podcast on the level of patient preparation for a doctor visit. Despite the small set of participant, the study assessed all relevant aspects of the topic.
    • Thank you very much for your comment. We are aware of the relatively small number of patients in our study. However, as this is a qualitative study aimed at understanding how patients experience the podcasts and information, we did not find this problematic. Any potential challenges related to the sample size are addressed in the Strengths and Limitations section.
  • Comment: The abstract lack the study results and conclusion.
    • We agree with the reviewer and have reorganized the abstract, making the different sections explicit. The results were included in the abstract, however we agree that they were not clearly presented. This have now been addressed (line 21-35).
  • Comment: It is confusing if the podcast is based on a previous diagnosis or only on the symptoms reported by the paptient self? The authors are advised to make this clear.
    • Thank you for your feedback. We have revised the content to clarify that three distinct podcasts were developed. “In April 2021, three user-centered podcasts [23–25], designed to prepare patients with suspected lung cancer, COPD, and sleep apnea for hospital visits, were introduced and made available for patient access. These podcasts were created through collaboration with patients undergoing diagnosis, ensuring their practical relevance. The podcasts were structured in a dialogue-based format, presenting conversations between patients and clinicians.” (line 114-120).
  • Comment: You need to put in brackets at some parts of papers  link to the results section. Many of my questions get answered in the discussion and results section. To make the paper easily readable, please the link pointed parts to the results and discussions.
    • We are not sure if we have understood this comment correctly. Many sections were highlighted by the reviewer, and it was not clear which ones needed links. Normally, you do not refer to a section within the same paper. We have tried our best to address each point, and hope it is acceptable in the revised version. If we still need to clarify further, please do let us know.
  • Comment: Mix of disease literacy and podcast related questions
    • The interview guide included a mix of disease literacy and podcast-related questions. This structure was intentional, with questions at the beginning and end designed to ease participants into the conversation and provide a comprehensive understanding of their experiences. By starting with disease literacy questions, we aimed to establish a baseline of the participants' knowledge and comfort level with their condition. This approach helped to build rapport and set the context for subsequent questions about the podcast. The concluding questions served to wrap up the discussion and ensure we captured any final thoughts or reflections on both the disease literacy and podcast-related topics.

We believe this structure allowed for a more natural flow of conversation and a deeper exploration of the participants' perspectives.

We added this to section 2.4: “The interview guide intentionally combined disease literacy and podcast-related questions. Questions at the beginning and at the end of the interview eased participants into the conversation for a more comfortable interview experience.” (line 141-143).

  • Comment: Two different diseases could have similar symptoms, based on that I would like to know how hopitals select the right postcast for the patient?
    • We have added this to section 2.3: “The podcast for each patient was chosen based on their referral from the practitioner doctor, ensuring it was tailored to their specific diagnosis”. (line 130-132).
  • Comment: Did you assess the techstress of these elders? In the literature, it is well known that elderlies develop technostress or technology resistence. Since you select this participants set, it will be important to assess their level of adherence to the podcast usage. The sample used for this study might biase the results, then you do not know what are mental and hearing health of these people
    • We thank the reviewer for pointing out these relevant questions regarding the impact of technology resistance and ‘tech-stress’ in the elderly population. It is indeed very relevant to study the association between tech-stress/technology resistance and podcast usage, however, that was outside the scope of the present study. Larger samples and other methodologies are needed for that purpose. We have addressed this in the discussion section (line 499-501).
  • Comment: How convenient is this for elderly?
    • We have included a discussion of age-related factors in podcasts throughout the discussion, especially in section 4.1 (line 388-427)
  • Comment: email or paper?
    • We agree that it could be more clear. Therefore we made a footnote* : “The letter was sent via e-boks, a Danish digital mailbox service”
  • Comment: The abstract should reflect what the contributes to.
    • We agree with the reviewer’s suggestion and have revised the abstract to better reflect the contributions of our study. We have added the following conclusion: “Podcasts can serve as a supplement to traditional information sources for patients. However, it is important to recognize that not all patients may be able to engage with this medium effectively due to technical challenges or personal preferences” (line 32-35).
  • Comment: Althrough that the sampling reflect the today society structure (from patient population with such diseases ), I think that the number (24) of people is quite small. It will be judiciouse to include jung people (18 to 40 J) too.
    • In line with qualitative methodology, it was not our aim – neither would it make sense – to include a sample of participants that are representative of the target population. We did not strive towards representativeness, but we strived towards a rich understanding of subjective experiences with podcasts in this particular context. There may be age-related differences, but it was outside the scope of the present study to address that particular question.  We have added this in section 4.3: “The group of participants in the present study included patients with suspected lung cancer, COPD, and sleep apnea, which are predominantly conditions seen in older individuals. Younger participants may have other experiences of podcasts in this context, but it was outside the scope of this particular study to address that research question.”